# Exploring the Inter-Monthly Dynamic Patterns of Chinese Urban Spatial Interaction Networks Based on Baidu Migration Data

Heping Jiang [1,2,3,4], Shijia Luo [1,2,3,4], Jiahui Qin [5], Ruihua Liu [1,2,3,4], Disheng Yi [1,2,3,4], Yusi Liu [1,2,3,4] and Jing Zhang [1,2,3,4,*]

1 College of Resources Environment and Tourism, Capital Normal University, Beijing 100048, China
2 Beijing Laboratory of Water Resources Security, Capital Normal University, Beijing 100048, China
3 3D Information Collection and Application Key Lab of Education Ministry, Capital Normal University, Beijing 100048, China
4 Beijing State Key Laboratory Incubation Base of Urban Environmental Processes and Digital Simulation, Capital Normal University, Beijing 100048, China
5 School of Environment and Spatial Informatics, China University of Mining and Technology, Xuzhou 221116, China
* Correspondence: zhangjings@mail.cnu.edu.cn; Tel.: +86-10-6890-2573

**Abstract:** The rapid development of the economy promotes the increasing of interactions between cities and forms complex networks. Many scholars have explored the structural characteristics of urban spatial interaction networks in China and have conducted spatio-temporal analyzes. However, scholars have mainly focused on the perspective of static networks and have not understood the dynamic spatial interaction patterns of Chinese cities. Therefore, this paper proposes a research framework to explore the urban dynamic spatial interaction patterns. Firstly, we establish a dynamic urban spatial interaction network according to monthly migration data. Then, the dynamic community detection algorithm, combined with the Louvain and Jaccard matching method, is used to obtain urban communities and their dynamic events. We construct event vectors for each urban community and use hierarchical clustering to cluster event vectors to obtain different types of spatial interaction patterns. Finally, we divide the urban dynamic interaction into three urban spatial interaction modes: fixed spatial interaction pattern, long-term spatial interaction pattern, and short-term spatial interaction pattern. According to the results, we find that the cities in well-developed areas (eastern China) and under-developed areas (northwestern China) mostly show fixed spatial interaction patterns and long-term spatial interaction patterns, while the cities in moderately developed areas (central and western China) often show short-term spatial interaction patterns. The research results and conclusions of this paper reveal the inter-monthly urban spatial interaction patterns in China, provide theoretical support for the policy making and development planning of urban agglomeration construction, and contribute to the coordinated development of national and regional cities.

**Keywords:** urban spatial interaction network; dynamic spatial interaction patterns; dynamic community detection; Baidu migration data

## 1. Introduction

In the process of urbanization in China, the rapid growth of the economy and the development of science and technology greatly reduce the cost in the exchange of various elements between cities. As a result, interactions between cities are becoming closer and more diverse. Castells proposed the theory of "space of flows" to understand the interaction of cities [1,2]. In the "space of flows" theory, the cities exist as the endpoint of the flow but do not create the urban network themselves. The essence of the interaction of the urban spatial interaction network are feature flows based on the urban infrastructure network, for

example, the population flow, information flow, and commodity flow [3]. Therefore, we can use these flow data to form urban spatial interaction networks. With the combination of network data and the GI system, we have better approaches to understand the network characteristics and spatial characteristics of spatial information networks [4–6]. Establishing an urban spatial interaction network is an appropriate way to explore the network structure characteristics and the spatial interaction patterns of cities. Many scholars put the attention on it under various perspectives [7]. From the global perspective, urban spatial interaction network research is conducive to providing a research basis and theoretical reference for the formulation of urban agglomeration development policies and development direction. In addition, under the perspective of an individual city, the results also help to clarify the development orientation of the city itself and formulate reasonable development strategies and planning.

In the era of big data, there are a lot of big geo-data. Many of these data carry spatial interaction information between many cities. Scholars have used a variety of data to build urban spatial interaction networks, such as traffic flow or cargo flow composed of aviation flight data, railway transport, highway, and shipping data [8–11]; capital flow data composed of company network distribution and enterprise relations [12–14]; and information flow data composed of Twitter, Weibo, mobile call numbers, etc. [15–18]. This series of flow data provides a variety of cases, which offer support for understanding urban spatial interaction networks, but these data are biased and can only provide part of the insights of interactions between cities. Population migration data has many advantages, such as the wide coverage of people and comprehensive transportation modes, and is the main driving force of other data flows [19]. Scholars have noticed networks of population migration data and have used social network analysis (SNA) to understand the migration structure [20]. Not only that, but researchers have also considered the spatial context of migration data, such as by using gravity models with spatial distance [21,22] and analyzing the rationality of administrative division through the spatial distribution of population flow [23]. The network characteristics and spatial context of population migration data have good applicability to the study of urban interaction. Therefore, studies on migration networks and urban spatial interaction networks established by population flow have emerged one after another.

In the study of urban interaction networks based on population migration, some scholars have focused on the relationship between the volume of migration and the properties and distance of the city itself. They fitted or predicted migration volume with the gravity model and the radiation model [24,25]. Other scholars have focused on the study of urban spatial interaction network structure. They [26–28] used indexes such as degree centrality, clustering coefficient [29], and PageRank [30] to understand the characteristics of urban spatial interaction networks and obtain the spatio-temporal interaction structure between cities. Pitoski integrated relevant studies on migration data and proved the applicability of these research methods [31]. Scholars have conducted a variety of spatial and temporal research. Lai et al. studied the urban interaction structure during the Spring Festival of 2016-2017 by using Tencent migration data from 346 cities in China [32]. Zhang et al. described the structure of the national urban network and the regional urban network under daily migration data [19]. In the study of inter-annual network structure, Xiang et al. analyzed the spatial differences and spatio-temporal changes in urban interaction in recent years by using the data of 2015–2019 inter-annual Spring Festival transport [33].

Many meaningful studies have been based on urban spatial interaction networks, but all of them were carried out from the perspective of static networks. Obviously, urban interaction is a dynamic process that changes over time. Although some scholars have analyzed the spatio-temporal characteristics of urban spatial interaction networks, such analyzes have failed to reveal the change processes of the networks. In addition, there have been few studies on inter-monthly urban interaction patterns. Therefore, the introduction of the dynamic network perspective into urban interaction networks can bring discoveries to urban spatial interaction patterns. In the dynamic urban spatial interaction network,

it changes over time. Nodes and edges can be added to or removed from the network. In weighted networks, weights can also evolve [34]. The dynamic community detection method can be used to explore the dynamic changes in nodes' spatial interaction patterns in networks. The results of dynamic community detection are communities composed of closely related nodes. As the nodes and edges in the dynamic network change with time, the communities also change, and these changes of the community are called dynamic events. Dynamic community detection has also been applied to various fields, such as science citation networks [35], social interaction analyzes [36,37], geographic interactions [38], brain functional research [39,40], etc. In the dynamic network of urban interaction, the urban communities and their dynamic events are the results of dynamic community detection. The urban communities represent the groups of cities that are more connected internally than externally, reflecting the urban agglomeration structure, as well as the dynamic events which represent the changes in the urban agglomeration structure. According to various events in the dynamic community (such as growth, contraction, merger, split, etc.) [41], the inter-monthly dynamic change mode of urban agglomeration can be understood.

In order to explore the monthly dynamic patterns of the Chinese urban spatial interaction network, this paper intended to use Baidu migration data from October 2020 to December 2021 to build monthly urban spatial interaction networks through data pre-processing. The dynamic community detection algorithm, combined with the Louvain algorithm and the Jaccard matching method, was used to mine dynamic urban communities and community dynamic events. Then, we used the hierarchical clustering algorithm to cluster all the existing dynamic communities, and the clustering results represented the monthly dynamic interaction patterns of Chinese cities. The contributions of this article are as follows:

- This paper proposed a new research framework for learning urban dynamic interaction, which used a dynamic community detection algorithm and a clustering algorithm to mine the urban dynamic interaction patterns.
- By using Baidu migration data, we learned the inter-monthly dynamic interaction patterns of Chinese cities.

The structure of this paper is as follows: Section 2 introduces the study area of this paper, the datasets, and the data pre-processing. Section 3 introduces the principle and application of the methods used in this paper. Section 4 shows the results of these methods. In Section 5, we have a further discussion of these research results and conclude the paper.

## 2. Study Area and Datasets

### 2.1. Study Area

China is a vast country, with a land area of 9.6 million square kilometers, ranking the third in the world, and a water area of more than 4.7 million square kilometers of inland sea and marginal sea. China has forty-four provincial-level administrative regions, including twenty-three provinces, five autonomous regions, four municipalities directly under the central government, and two special administrative regions. China also has 333 prefecture-level cities [42]. In Figure 1, we show the administrative divisions at the municipal level and above in China, which were used as the research areas of this study.

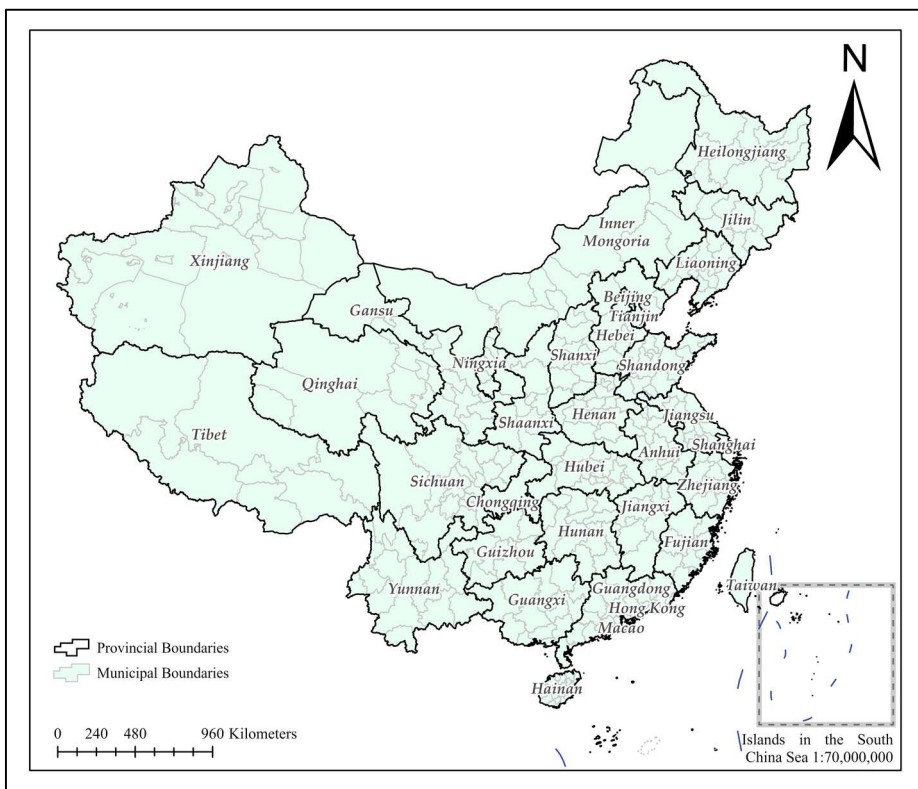

**Figure 1.** Study area. Administrative divisions of provinces and cities in China.

### 2.2. Datasets and Data Pre-Processing

The main dataset used in this paper is Baidu migration data, one of Baidu Maps' LBS (location based on service) datasets. Baidu Maps is one of the largest providers of map content, navigation, and location services in China. Baidu Maps records users' location and movement information to generate migration data. Baidu Maps generates migration data if a user temporarily leaves the city where he or she lives for a long time and moves to another city. We obtained Baidu migration data by posting HTML (Hyper Text Markup Language) requests to the Baidu Migration platform [43]. However, for protecting users' privacy, the Baidu Migration platform only records the percentage of people moving daily in and out of 368 prefecture-level and county-level cities (excluding Taiwan) (e.g., in data of people from Shanghai moving into Beijing on a certain day, we can only know the proportion of the migrating population from Shanghai to Beijing from the total migrating population, but not the specific moving population). After data collection, we had total migration data of 368 cities for 457 days from 1 October 2020 to 31 December 2021. The monthly snapshot of the dynamic network was established by using these data, and the data pre-processing process was as follows:

1.  Spatial interaction strength definition.

Determining the interaction intensity between different cities is the premise of constructing an urban interaction network. To construct the monthly dynamic network, we calculated the average of the monthly average percentage of immigration and emigration of two cities to obtain the spatial interaction strength of two cities in a month. The calculation equation of spatial interaction strength was as follows:

$$S_{i,j} = \left( \overline{P}_{i,j}^{out} / \overline{P}_i^{out} + \overline{P}_{i,j}^{in} / \overline{P}_j^{in} \right) / 2 \tag{1}$$

where $S_{i,j}$ represented the spatial interaction strength of city $i$ and city $j$. $\overline{P}_{i,j}^{out}$ represented the monthly average emigration from city $i$ to city $j$. $\overline{P}_i^{out}$ represented the total monthly average

emigration of city *i*. $\overline{S}_{i,j}^{in}$ represented the monthly average immigration from city *i* to city *j*. $\overline{P}_{j}^{in}$ represented the total immigration of city *j*. Finally, we obtained a city adjacency matrix for one month. The matrix was a symmetric matrix with the main diagonal of zeros, and the elements in the matrix represented the strength of the connection between two cities.

$$
\begin{pmatrix}
0 & S_{1,2} & \cdots & S_{1,j} \\
S_{2,1} & 0 & \cdots & S_{2,j} \\
\vdots & \vdots & \ddots & \vdots \\
S_{i,1} & S_{i,2} & \cdots & 0
\end{pmatrix}
\tag{2}
$$

2.    Constructing Dynamic Urban Spatial Interaction Networks.

After the spatial interaction strength definition, we derived the city nodes table including the names of the city nodes, locations of the city nodes (longitude and latitude), and the intensities of the interactions. The sample data are shown in Table 1. We used the city nodes table and the city adjacency matrix to construct urban spatial interaction networks. To reduce the influence of edges with low interaction strength, we removed the edges with interaction strength less than one, and established a weighted undirected graph for each month. From October 2020 to December 2021, there were 15 city spatial interaction networks with the same nodes in each network and a total of 368 city nodes.

**Table 1.** The city nodes table of the sample data.

| City A | Location A | City B | Location B | Spatial Interaction Strength | Month |
|--------|-----------|--------|-----------|------------------------------|-------|
| Beijing | 116.40, 34.90 | Tianjin | 117.20, 39.08 | 24.38 | 2021 Jan. |
| Beijing | 116.40, 34.90 | Shijiazhuang | 114.51, 38.04 | 6.08 | 2021 Jan. |
| Tianjin | 117.20, 39.08 | Shijiazhuang | 114.51, 38.04 | 3.95 | 2021 Jan. |
| ... | ... | ... | ... | ... | ... |

## 3. Methodology

The technical route of this study is shown in Figure 2 below. According to Figure 2, our work was divided into three parts: data pre-processing, dynamic community detection, and urban community analysis. In Section 2.2, we introduced the data pre-processing and construction process of the urban spatial interaction networks in detail. After that, the research methods we used were mainly in the last two parts: in dynamic community detection, we used the Louvain algorithm to obtain city communities and the Jaccard matching method to derive their change events from the urban spatial interaction network. In the urban communities analysis, we build event vectors for each urban community and used the hierarchical algorithm to cluster event vectors. Finally, we summarized the dynamic change in urban communities into several patterns. Here, the methods used in this paper are elaborated.

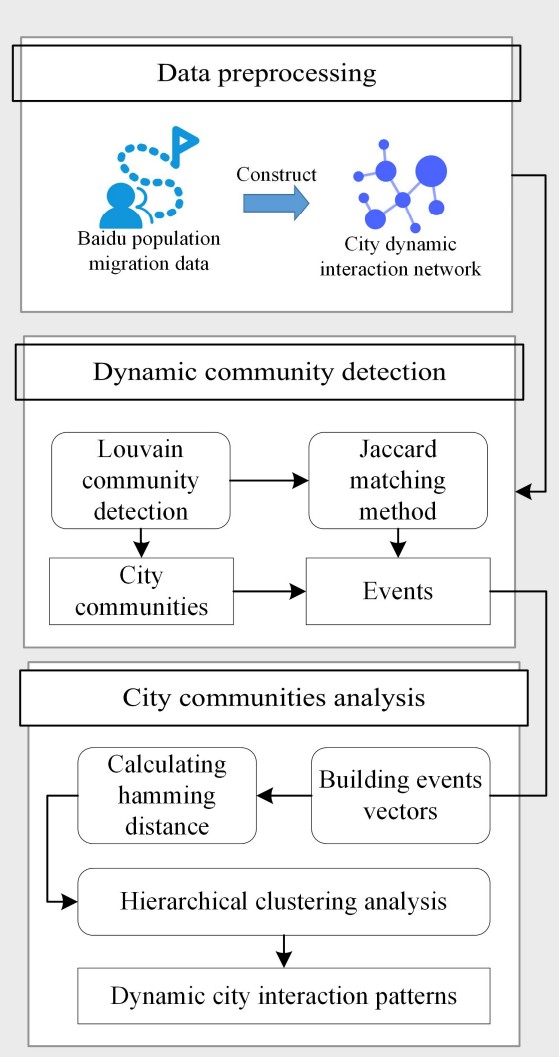

**Figure 2.** The technical route diagram.

### 3.1. Dynamic Community Detection

At present, dynamic community detection algorithms are mainly divided into three types: instant optimal community detection, temporal trade-off community detection, and cross-time community detection [44]. Among them, the instant optimal dynamic community detection algorithm was based on the well-studied static network community detection algorithm and obtained community and community change events at each moment. Based on the instant optimal community detection strategy, we combined the Louvain static community detection algorithm and the Jaccard matching method as the dynamic community detection algorithm.

#### 3.1.1. Louvain Community Detection Algorithm

The Louvain algorithm is one of the most widely used community detection algorithms at present. It was proposed by Blondel et al. in 2008 [45]. The algorithm is based on the idea of greed, aiming at modularity optimization. Modularity was a common indicator to evaluate the partition result of a community in the undirected weighted graph [46,47], as shown in Equation (3).

$$Q = \frac{1}{2m} \sum_{i,j} \left[ S_{ij} - \frac{k_i k_j}{2m} \right] \delta(c_i, c_j) \tag{3}$$

where Q was the modularity, $S_{ij}$ represented the weight between node $i$ and node $j$, $k_i$ and $k_j$ were the sums of the weights of the edges attached to node $i$ and node $j$, and $m$ was equal to half of the sum of all the edge weights. $\delta(c_i, c_j)$ was a binary function; the result was 1 if $c_i = c_j$ and 0 otherwise. The whole algorithm was divided into two steps, and the process of the algorithm is shown in Figure 3.

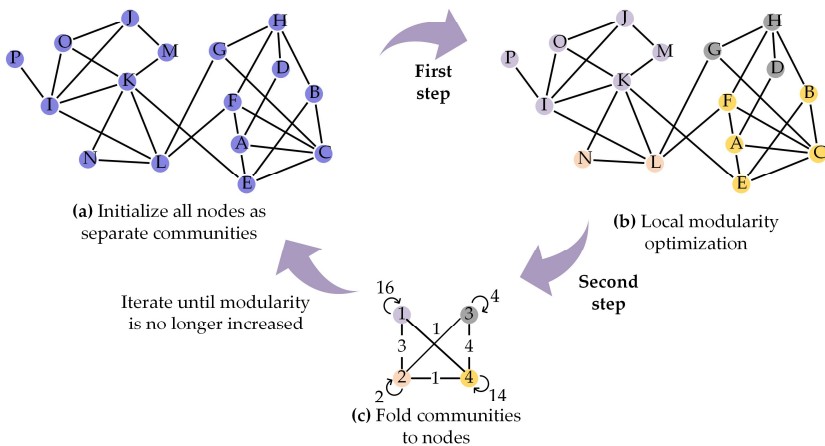

(a) Initialize all nodes as separate communities

**First step**

(b) Local modularity optimization

Iterate until modularity is no longer increased

**Second step**

(c) Fold communities to nodes

**Figure 3.** The process of the Louvain algorithm. (**a**) is the initial setup of the network, assigning all nodes as separate communities, (**b**) is the result of local modularity optimization, where different colors mean different communities, and (**c**) is the result of folding communities into new nodes, forming a new network, where the new network contains edges not only between nodes but also within nodes.

1. **First step:** local modularity optimization.

At first, we assigned each node to a separate community, as in Figure 3a) Then, the algorithm scanned all nodes in the network, traversed all the neighbor nodes of each node, and measured the gain in modularity brought by adding the node to its neighbor's community. In addition, we selected the neighbor node with a maximum gain greater than 0 to join its community. This process was repeated until the community belonging to each node was not changed, as in Figure 3b. The gain in modularity from node $i$ to community $C$ was calculated by Equation (4).

$$\Delta Q = \left[ \frac{\sum_{in} + 2k_{i,in}}{2m} - \left( \frac{\sum_{tot} + k_i}{2m} \right)^2 \right] - \left[ \frac{\sum_{in}}{2m} - \left( \frac{\sum_{tot}}{2m} \right)^2 - \left( \frac{k_i}{2m} \right)^2 \right] \quad (4)$$

where $\sum_{in}$ was the sum of the weights of edges inside $C$, $k_{i,in}$ was the sum of the weights of edges from node $i$ to nodes in $C$, $\sum_{tot}$ was the sum of the weights of edges that connected to the nodes in $C$, and other variables were the same as in Equation (3).

2. **Second step:** folding the communities into nodes.

We folded each community formed in the first step. Then, we calculated the weights of the edges in the newly generated nodes and the sum of the weights between newly generated nodes, respectively, as in Figure 3c.

These two steps were repeated until the modularity no longer increased. Finally, the final partition result was the calculation result of the algorithm.

### 3.1.2. Jaccard Matching Method

The Jaccard matching method is based on the Jaccard score. We could understand the change events between communities by judging the Jaccard score of communities in different months. The Jaccard score, also called intersection over union, is a statistic used to compare the similarity and diversity of sets. The Jaccard score can measure the similarity of

finite sample sets and is defined as the proportion between the intersection size and union size of two sets [48,49]. The calculation formula of the Jaccard score was Equation (5):

$$J(A, B) = \frac{|A \cap B|}{|A \cup B|} = \frac{|A \cap B|}{|A| + |B| - |A \cap B|} \tag{5}$$

In the formula, $J(A, B)$ meant to calculate the Jaccard score of community $A$ and community $B$; this value ranged from 0 to 1. The larger the value of the Jaccard score, the more identical elements there were in the two communities. In this study, the higher the Jaccard score of urban communities between two adjacent months was, the closer the relationship between the two urban communities was. If the Jaccard score was 0, it represented that there was no relationship between the two urban communities. In the dynamic urban spatial interaction network, city nodes did not disappear or appear. In addition, the emergence or disappearance of urban communities was the result of the split or merger of previous urban communities. Therefore, this paper only set the following 5 event types [34].

- **Growth:** a community that grew by integrating new city nodes.
- **Contraction:** a community that contracted by rejecting some of its city nodes.
- **Merge:** two communities or more that merged into a single one.
- **Split:** one community that split into two or more communities.
- **Continue:** a community that did not change.

According to these event types, we defined events for the change in urban communities according to the results of the Jaccard score of urban communities in different months, as shown in Table 2. Setting thresholds was important for defining the events. In this study, the distribution of the Jaccard score was used to determine the specific threshold.

**Table 2.** Event definition table. $A$ is the urban community of the month before and $B$ is the urban community of the month after.

| Jaccard Score | Relationship | Events |
|:---:|:---:|:---:|
| $J(A, B) = 1$ | $A = B$ | Continue |
| Threshold $\leq J(A, B) < 1$ | $A \subseteq B$ <br> $A \supseteq B$ | Growth <br> Contraction |
| $0 \leq J(A, B) <$ threshold | $A \subseteq B$ <br> $A \supseteq B$ | Merge <br> Split |
| 0 | - | No event |

### 3.2. Hierarchical Clustering Method

After dynamic community detection, we obtained the results of urban communities and their dynamic events for each month. These urban communities represented urban agglomerations. Although the dynamic interaction patterns in different urban agglomerations were different, we could still cluster the types of interaction patterns of urban communities according to their change patterns over time. Therefore, the events of urban agglomerations were constituted into event vectors, the sample vector is shown in Equation (6).

$$\{\text{event}(2020 \text{ Oct.}\text{-}2020 \text{ Nov.}), \text{ event}(2020 \text{ Nov.}\text{-}2020 \text{ Dec.}) \cdots, \text{ event}(2021 \text{ Nov.}\text{-}2021 \text{ Dec.})\} \tag{6}$$

Then, we adapted the hierarchical clustering method to cluster event vectors. Hierarchical clustering is a method of cluster analysis that seeks to build a hierarchy of clusters. The strategies for hierarchical clustering generally fall into two types: bottom-up aggregation and top-down division [50]. The results of hierarchical clustering are usually presented in a dendrogram [51]. Thus, the results of hierarchical clustering are more intuitive than those of other clustering methods. The hierarchical clustering method includes a variety of distance measure metrics. Since the elements in the event vector were not specific values,

this paper used the Hamming distance as the distance function of the event vector [52]. The Hamming distance between the two event vectors was the number of different events at the corresponding positions of the two vectors, and the equation was as follows:

$$D_{hamming} = \sum_{i}^{n}(X[i] \oplus Y[i]) \tag{7}$$

where X and $Y$ were the two urban communities' event vectors, n was the length of the event vectors, $X[i] \oplus Y[i]$ was the XOR operation (if $X[i]$ and $Y[i]$ are the same, return 0, otherwise return 1). As we could see, the larger the Hamming distance was, the more differences between the two event vectors there were. Hierarchical clustering was performed according to the Hamming distance between the event vectors, and a clustering dendrogram was obtained. In order to determine the reasonable clustering level, we used the silhouette coefficient as the evaluation index of clustering results. The silhouette coefficient is a method that reflects the consistency of data clustering results and can be used to evaluate the degree of dispersion between clusters after clustering [53]. The equation was as follows:

$$S(i) = \frac{b(i) - a(i)}{\max\{a(i), b(i)\}} \tag{8}$$

where $a(i)$ was the average Hamming distance between event vector $i$ and other event vectors in the same cluster and $b(i)$ was the average Hamming distance between event vector $i$ and other event vectors in different clusters. We usually choose the cluster number with the highest silhouette coefficient as the best clustering result. These clusters represented different types of dynamic patterns of urban communities.

## 4. Results

In this section, we detailed the results of dynamic community detection, that is, dynamic communities and community events. The dynamic interaction model of the city was reflected by the event vectors clustering of the community.

### 4.1. Urban Communities and Dynamic Events

We used the dynamic community detection algorithm for the monthly urban spatial interaction network. The urban communities in each month were obtained, as shown in Figure 4.

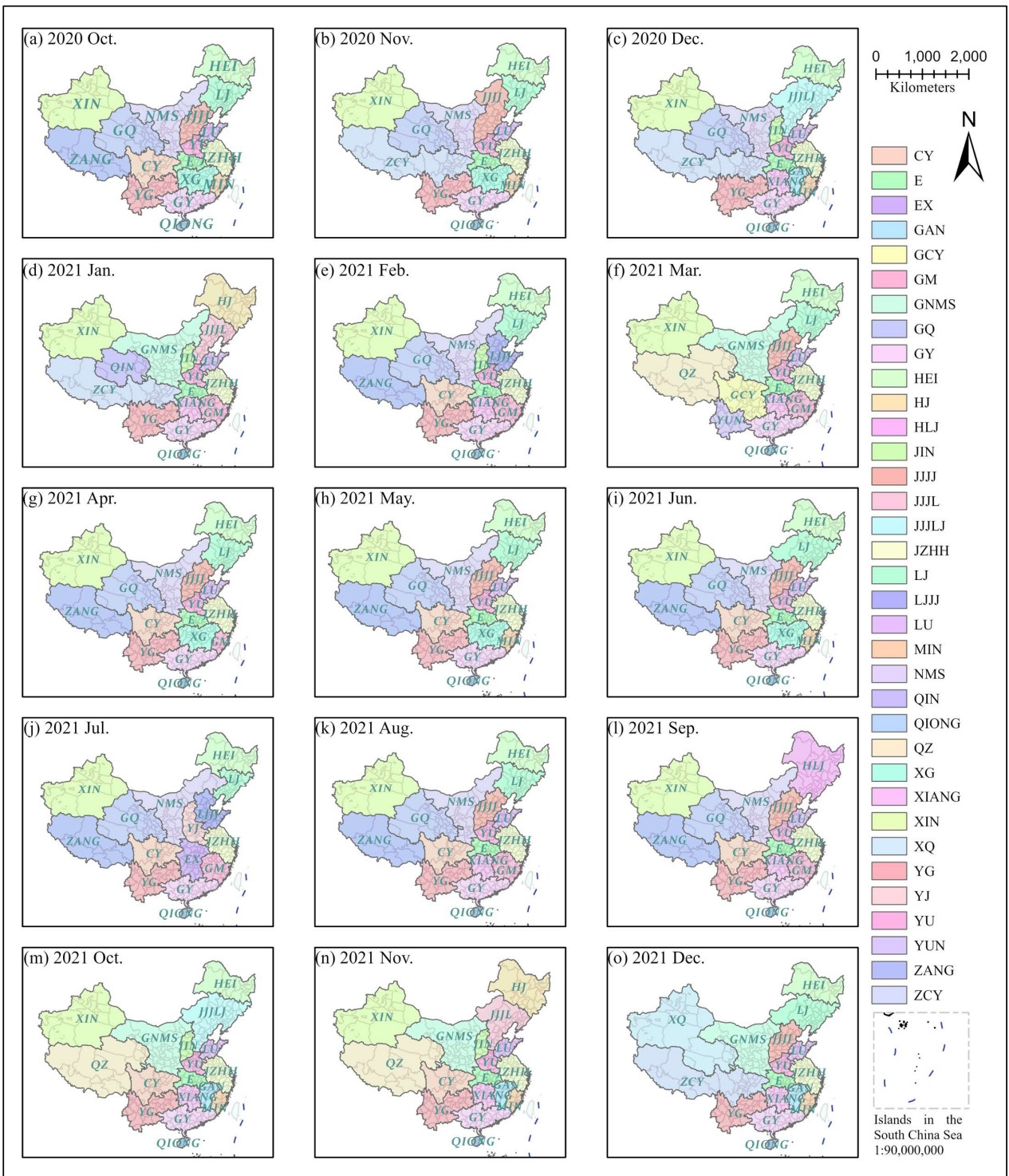

**Figure 4.** The urban communities of each month. Each sub-plot represents the information of the urban communities in that month.

According to Figure 4, there were 35 urban communities during the 15 months from October 2020 to December 2021. Because cities within a province were usually more closely connected, the coverage of the urban communities was usually composed of cities in one or more provinces. Thus, the urban communities were named by the abbreviation of the provinces which the communities contained. Table 3 shows the names of the urban communities and corresponding provinces, and readers can learn more about the provinces from Figure 1.

**Table 3.** The urban communities information table.

| Communities | Provinces | Communities | Provinces | Communities | Provinces |
|---|---|---|---|---|---|
| EX | Hubei; Hunan | JJJL | Beijing; Tianjin; Hebei; Liaoning | XIANG | Hunan |
| GAN | Jiangxi | JJJLJ | Beijing; Tianjin; Hebei; Liaoning; Jilin | XIN | Xinjiang |
| GCY | Guizhou; Sichuan; Chongqing | JZHH | Jiangsu; Zhejiang; Shanghai; Anhui | XQ | Xinjiang; Qinghai |
| GM | Jiangxi; Fujian | LJ | Liaoning; Jilin | YG | Yunan; Guizhou |
| GNMS | Gansu; Ningxia; Inner Mongolia; Shaanxi | LJJJ | Shandong; Beijing; Tianjin; Heibei | YJ | Henan; Shanxi |
| GQ | Gansu; Qinghai | LU | Shandong | YU | Henan |
| GY | Guangxi; Guangdong | MIN | Fujian | YUN | Yunnan |
| HEI | Heilongjiang | NMS | Ningxia; Inner Mongolia; Shaanxi | ZANG | Tibet |
| HJ | Heilongjiang; Jilin | QIN | Qinghai | ZCY | Tibet; Sichuan; Chongqing |
| HLJ | Heilongjiang; Jilin; Liaoning | QIONG | Hainan | XIANG | Hunan |
| EX | Hubei; Hunan | JJJL | Beijing; Tianjin; Hebei; Liaoning | XIN | Xinjiang |
| GAN | Jiangxi | JJJLJ | Beijing; Tianjin; Hebei; Liaoning; Jilin | | |

These urban communities represented the urban agglomeration structure formed by the interaction between cities in a certain month. As we could see, some urban communities stayed the same from month to month (continue event), some urban communities changed the number of cities they contained (growth or contraction event), and some disappeared or appeared (split or merge event). Thus, events were formed by the monthly dynamics of these urban communities. To define the dynamic events of the communities, we had to define the thresholds in Table 2 beforehand. We calculated the distribution of the results of the Jaccard scores in Figure 5.

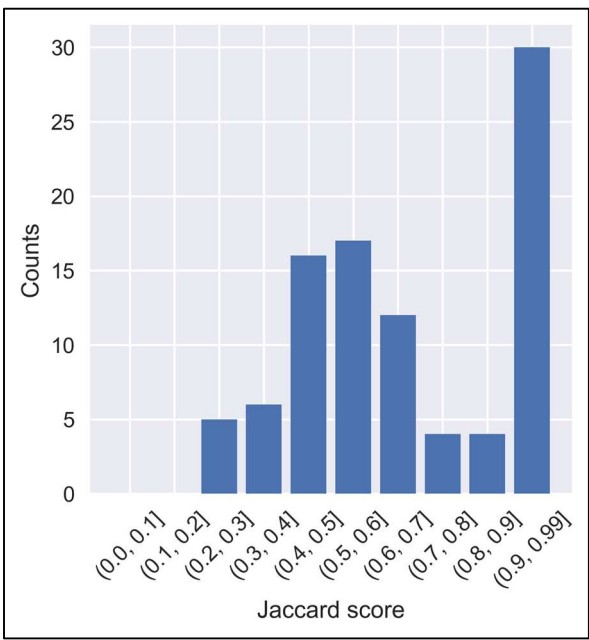

**Figure 5.** The distribution of Jaccard scores.

This threshold was used to distinguish between growth, contraction, split, and merge events. A high value of the Jaccard score indicated growth or contraction, while a low value indicated split or merging. According to the distribution in Figure 5, we set the threshold at 0.8 to ensure the balance of the number of dynamic events.

According to Table 2 and the threshold of 0.8, we defined the dynamic events. We calculated statistics on the dynamic events of urban communities in each period, and the results are shown in Figure 6.

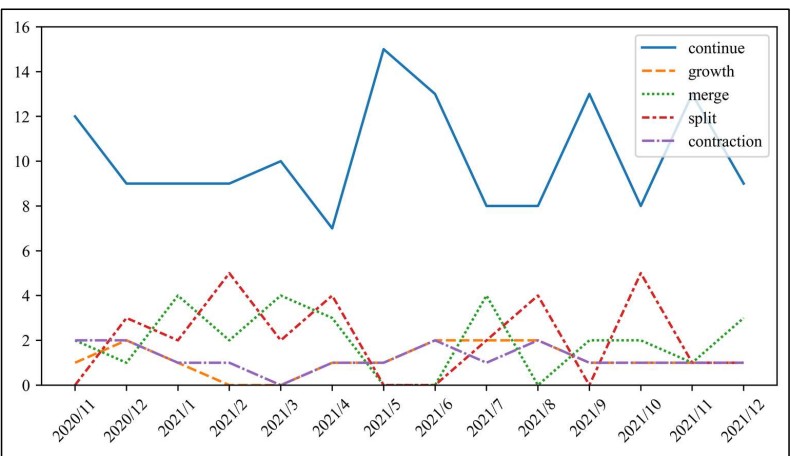

**Figure 6.** The numbers of five dynamic events in each period.

In this figure, the vertical axis shows the number of events that occurred. The horizontal axis shows the period in which they happened (e.g., 2020/11 means the period from October to November 2020). From Figure 6, we obtained an overview of the dynamics of these urban interactions. Continuous events dominated over all periods, but other events still existed in each period, which indicated that the inter-monthly dynamic of interactions in most cities was stable, and only a few cities changed their interaction states. Besides the continue events, the high-frequency events were the split and merge events, and the low-frequency events were the growth and contraction events. This phenomenon showed that the dynamic change in urban interaction often manifested as the merger and separation of urban agglomerations; this was mainly because most cities highly interacted within the urban agglomeration, and few cities were transferred from one urban agglomeration to another. Thus, the number of merge and split events was more than the growth and contraction events. Here, it was only a preliminary description of the results of dynamic community detection. Next, we further studied the clustering results of these urban community event vectors and explored the dynamic patterns of these urban agglomerations.

### 4.2. Dynamic Patterns of Urban Agglomeration

In Section 4.1, we derived the urban communities and their dynamic events and carried out the overall statistical analysis. To further explore the dynamic interaction patterns of these urban communities, we constructed event vectors for each urban community and performed hierarchical clustering. Figure 7 shows the dendrogram of hierarchical clustering and the Hamming distance between different urban communities.

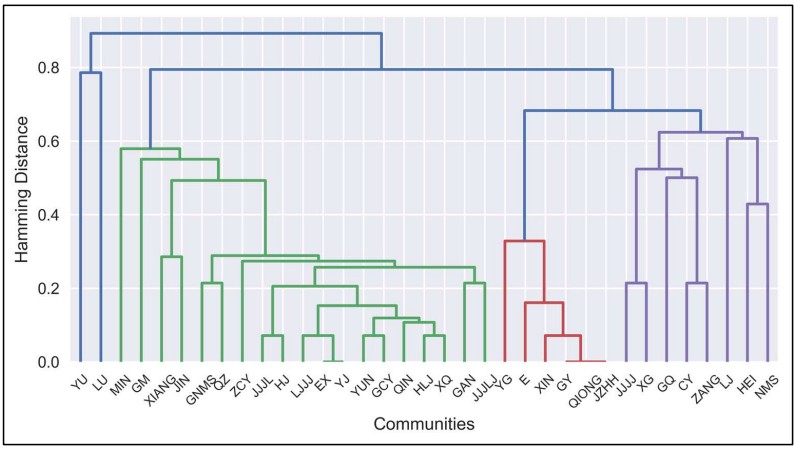

**Figure 7.** The dendrogram of the hierarchical clustering.

Through this dendrogram, we intuitively learnt the distance relationship of each event vector. For finding the appropriate number of clusters, we used the silhouette coefficient, and the result is shown in Figure 8.

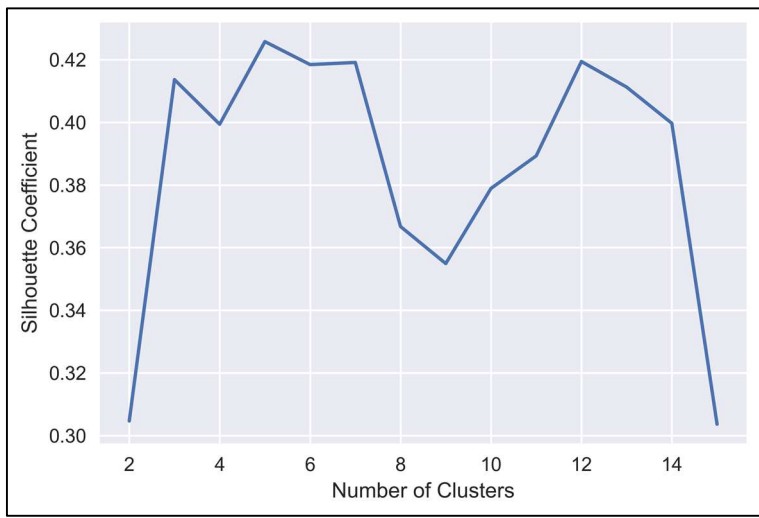

**Figure 8.** The silhouette coefficient under the different numbers of clusters.

According to Figure 8, when the number of clusters was five, the silhouette coefficient had a maximum value of 0.426. Therefore, we set the number of clusters to five. The urban communities contained in these five clusters are shown in Table 4.

**Table 4.** Clustering results of urban communities.

| Clusters | Urban Communities |
| :---: | :---: |
| C0 | QIONG; JZHH; GY; YG; XIN; E |
| C1 | HEI; CY; LJ; GQ; NMS; ZANG; JJJJ; XG |
| C2 | XIANG; MIN; GM; JIN; GNMS; ZCY; GAN; QZ; LJJJ; JJJL; HJ; JJJLJ; EX; QIN; HLJ; YUN; XQ; YJ; GCY |
| C3 | YU |
| C4 | LU |

For understanding the dynamic pattern types of urban communities under different clusters, we counted the number of periods each urban community existed. The number of times urban communities existed is shown in Figure 9.

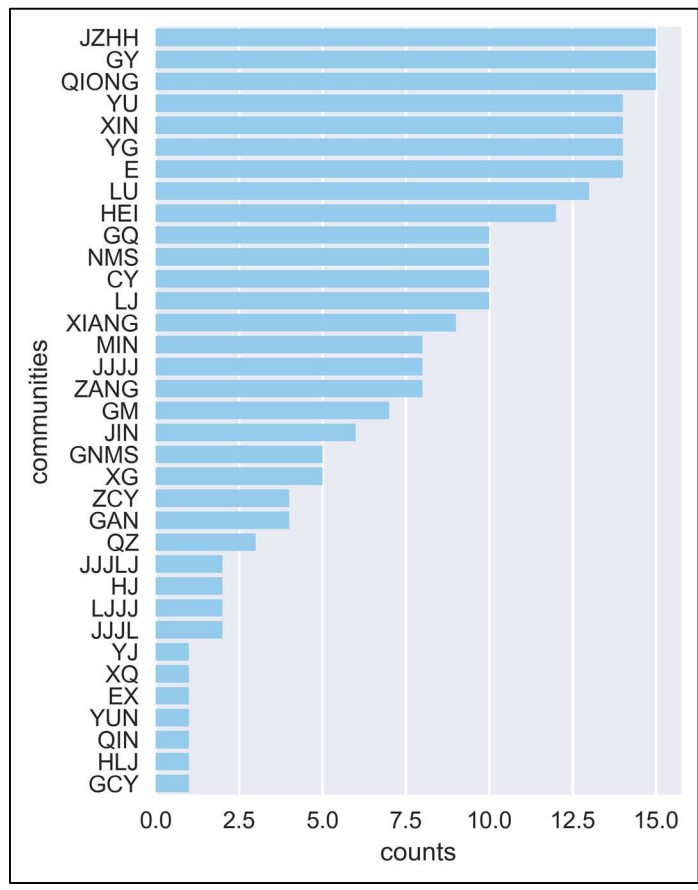

**Figure 9.** Number of times urban communities existed.

By using information about the existence frequency of urban communities and the clustering results, we divided the dynamic changes in these urban communities into three dynamic spatial interaction patterns. We called these three dynamic spatial interaction patterns fixed spatial interaction pattern, long-term spatial interaction pattern, and short-term spatial interaction pattern.

1.  Fixed spatial interaction pattern

The fixed spatial interaction pattern referred to the urban agglomeration with a very stable interaction state. It was seen from Table 4 and Figure 9, that the urban community in C0 existed for a long time and was characterized by stability. Due to the influence of economic conditions, transportation, and other factors, there were both economically developed urban communities and poorly developed urban communities with fixed spatial interaction patterns. Cities in the Yangtze River Delta and cities in the Pearl River Delta had good regional advantages, which drove the economic development of surrounding cities to form a stable dynamic structure, such as JZHH and GY. For the less developed northwest region, the influence of natural conditions and inconvenient transportation made it difficult for urban agglomerations in these regions to connect with the other cities, so a stable urban interaction structure formed, such as QIONG and XIN.

2.  Long-term spatial interaction pattern

The long-term spatial interaction pattern referred to the fact that the interaction of the city was in a stable state in most periods but changed in some special periods. In C1, all the urban communities had a high frequency of existence and were characterized by long-term interactions. In these urban communities, there were many typical urban agglomerations in central and western China, such as CY, YG, and NMS. The development levels of these cities were slightly lower than that of eastern urban agglomerations, but they

had a higher influence in central and western regions. In some periods, they attracted urban agglomerations in less developed areas to form new urban communities, such as in the merger of CY and ZANG to form ZCY, and the merger of NMS and GAN to form GNMS.

3.   Short-term spatial interaction pattern

The short-term spatial interaction pattern was a temporary urban interaction structure. The urban communities in C2 belonged to short-term spatial interaction patterns. The number of urban communities in this spatial interaction pattern was very large, and these urban communities manifested by the split and merge results of the urban communities in long-term spatial interaction patterns. Therefore, the urban communities of short-term spatial interaction patterns were more complex and diversified.

Finally, we found that YU and LU in C3 and C4 were two spatial cases. Puyang city was located at the junction of LU and YU. It had a close connection with both two urban agglomerations. Therefore, the events of these two urban communities often focused on growth and contraction and were very different from other urban community events. YU and LU were clustered into individual clusters. However, the growth or contraction event had little influence on the overall interaction structure of urban agglomerations. Hence, we considered that LU and YU also belonged to the long-term spatial interaction pattern.

## 5. Discussion and Conclusions

In the paper, we obtained the spatial interaction patterns of urban communities. To make the results easy to understand, we visualized the change times in urban community affiliation for each city, as shown in Figure 10.

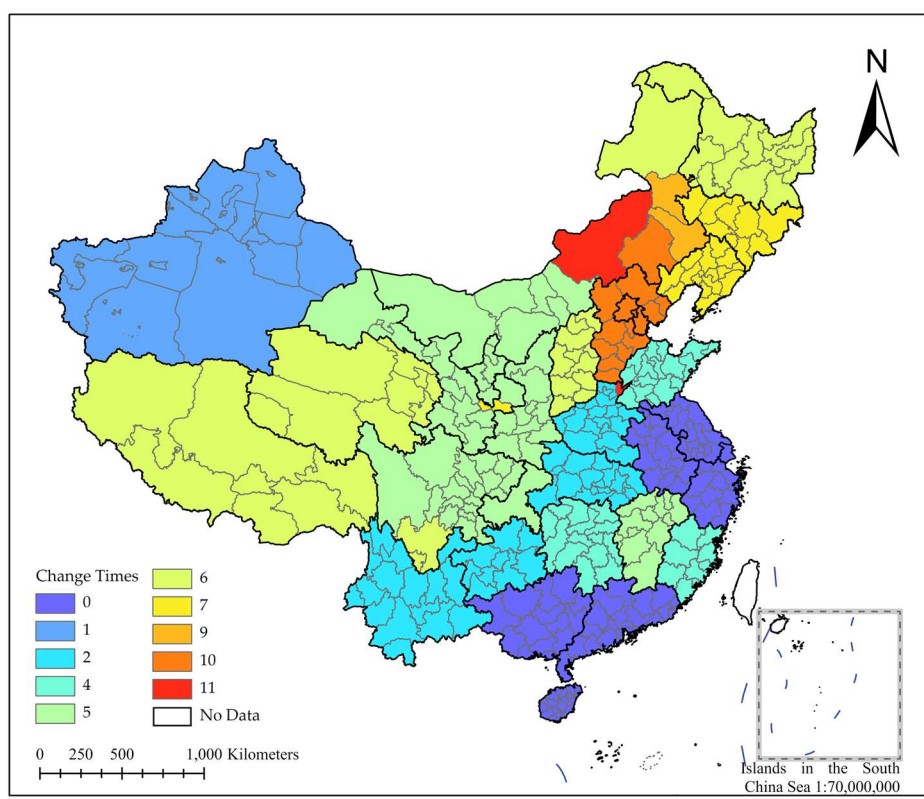

**Figure 10.** The change times in urban community affiliation for each city.

As seen in Figure 10, cities with fewer changes tended to belong to urban communities with fixed or long-term spatial interaction patterns. However, cities with more changes belonged to short-term spatial interaction patterns. In addition, we also noted that not all cities belonging to a province had the same number of changes. Based on the results and Figure 10, we summarized some interesting findings:

- The interaction between some cities on the edges of the provinces and cities of neighboring provinces was usually higher than those within the provinces, such as Hulunbeier and Chifeng in Inner Mongolia. There were also cities on the edge of the provinces that had strong interactions with both the cities within the provinces and adjacent provinces, such as Puyang in Henan.
- Some provinces that the public thought were highly interactive seemed to be not closely connected in the urban spatial interaction network. For example, the "three provinces in northeastern China" (Heilongjiang, Jilin, and Liaoning), "YunGuiChuan" (Yunnan, Guizhou, and Sichuan), and "Qinghai-Tibet Region" (Tibet and Qinghai), often mentioned by people, had not formed long-term spatial interaction patterns. Some provinces with large economic differences had formed fixed spatial interaction patterns instead, such as GY. Maybe they shared the same regional culture.

In this study, we used Baidu migration data to explore dynamic urban spatial interaction patterns and divided urban spatial interaction patterns into three types: fixed spatial interaction pattern, long-term spatial interaction pattern, and short-term spatial interaction pattern. Based on these results and above findings, we drew the following conclusions:

- Some cities both in developed and less developed areas showed relatively stable urban interaction structures. However, the reasons for their formation were different. Due to the radiation effect of big cities, economically developed areas interacted stably with surrounding cities to form independent communities. However, in the less developed regions, due to the limitations of geographical, economic, or traffic conditions, these cities did not interact with external cities and formed independent urban agglomerations themselves.
- The monthly dynamic changes in cities in medium-level developed areas were obvious. These cities were in central and western China. The radiation capacity of these cities was limited and only attracted other cities for a few periods. Therefore, the interaction of these cities tended to split and merge in different periods, representing short-term spatial interaction patterns.

Overall, the research framework of this paper provided a new idea for studying the dynamic interaction between cities, and these research results and conclusions were also beneficial to theoretical references for the formulation of urban and urban agglomeration development planning. For example, for urban agglomerations with short-term spatial interaction patterns in central China, the construction of urban agglomerations should be strengthened. For the urban agglomerations with fixed interaction patterns in the underdeveloped areas of northwest China, the transportation disadvantage should be remedied and economic activities with the central cities should be strengthened.

Although the research results of this paper explored the dynamic interaction patterns of Chinese cities, there were still some deficiencies.

- **Data**: The long duration and wide coverage of Baidu migration data have indeed played an important role in mining the inter-monthly dynamic patterns of Chinese cities. However, due to the protection of user privacy, we could not know the specific migration volume, which limited the further study of this paper.
- **Analysis:** Some "Gordian knots" in spatial interaction networks still exist [54]. Current visualization methods make it difficult to show the dynamic changes in urban spatial interaction. In addition, there are a lack of relevant spatial analysis methods to understand the driving force of dynamic change in urban inter-monthly interactions. It is mainly because urban interaction is affected by many factors, such as urban economic conditions, relevant policies of local governments, natural conditions of different regions, etc. Due to the lack of monthly statistical data on the urban economy, further analysis was not accessible.

Therefore, in future studies, we hope to add more data to explore the influencing factors or mechanisms of urban inter-monthly interaction dynamics. At the same time, it is

also an important direction of future research to integrate multiple flow data to understand urban spatial interaction networks more comprehensively.

**Author Contributions:** Conceptualization, Jing Zhang and Heping Jiang; methodology, Heping Jiang and Shijia Luo; data curation, Heping Jiang and Ruihua Liu; writing—original draft preparation, Heping Jiang; writing—review and editing, Jiahui Qin; Disheng Yi and Yusi Liu; visualization, Heping Jiang and Shijia Luo; supervision, Jing Zhang; All authors have read and agreed to the published version of the manuscript.

**Funding:** This research was funded by the National Nature Science Foundation of China (Grant number No. 42071376); this research was funded by the Open Project Program of the State Key Laboratory of Virtual Reality Technology and Systems, Beihang University (Grant No. 01122220010028).

**Institutional Review Board Statement:** Not applicable.

**Informed Consent Statement:** Not applicable.

**Data Availability Statement:** Not applicable.

**Conflicts of Interest:** The authors declare no conflict of interest.

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
