# Peer review of "Exploring the Inter-Monthly Dynamic Patterns of Chinese Urban Spatial Interaction Networks Based on Baidu Migration Data"

_ijgi, doi:10.3390/ijgi11090486_

Round 1

Reviewer 1 Report

While this paper makes an important contribution in understanding migration using social network analysis, there are three shortcomings that need to be addressed before this paper can be considered for publication.  

1. There has been significant work on using social networks to understand migration outside of China. In fact, there has been generic, broader work on human interactions that need to be cited in this journal article, in particular to create a background as to why the migration data is useful to understand human interactions.  Specific examples include:

Ratti, C., Sobolevsky, S., Calabrese, F., Andris, C., Reades, J., Martino, M., Claxton, R. and Strogatz, S.H., 2010. Redrawing the map of Great Britain from a network of human interactions. PloS one, 5(12), p.e14248.

Bilecen, BaÅŸak, Markus Gamper, and Miranda J. Lubbers. "The missing link: Social network analysis in migration and transnationalism." Social Networks 53 (2018): 1-3.

Tranos, E., Gheasi, M. and Nijkamp, P., 2015. International migration: a global complex network. Environment and Planning B: Planning and Design, 42(1), pp.4-22.

2. Since this paper has been submitted to a journal that emphasizes Geo-Information, papers that include an understanding of the "spatial aspects" of Social networks should also be cited.  For example, the following papers have further details on linking spatial data to social network analysis:

Andris, C., 2016. Integrating social network data into GISystems. International Journal of Geographical Information Science, 8816 (March), 1–23. doi:10.1080/13658816.2016.1153103.

Radil, S.M., Flint, C., and Tita, G.E., 2010. Spatializing social networks: using social network analysis to investigate geographies of gang rivalry, territoriality, and violence in Los Angeles. Annals of the Association of American Geographers, 100 (2), 307–326. doi:10.1080/00045600903550428.

Sarkar, D., et al., 2019. Metrics for characterizing network structure and node importance in spatial social networks. International Journal of Geographical Information Science, 33 (5), 1017–1039. doi:10.1080/13658816.2019.1567736

3. Finally a critical analysis of the use of migration data can be articulated by considering the following journal article:

Andris, C., Liu, X., and Ferreira, J., 2018. Challenges for social flows. Computers, Environment and Urban Systems, 70 (October 2017), 197–207. doi:10.1016/j.compenvurbsys.2018.03.008

Apart from the suggestions above, the methods are clear and relevant.  If an additional parameter could be developed to consider distance, it may make the paper more interesting.  Overall, I think this will be a very interesting and relevant paper once the changes suggested have been implemented.

Reviewer 2 Report

Well written, interesting, and thoroughly explained article.

I only detect an unnecessary repetition of method and results in the conclusions (lines 420-450): I would rather dedicate more comments on the research outlook.

Reviewer 3 Report

The article is a very interesting attempt to capture the spatial dynamics of interaction in China's urban network. The proposed algorithm is, of course, only one way of achieving the stated goal and I will not discuss its validity here. However, it is worth pointing out here some suggestions that will certainly make the whole paper a little more readable which will facilitate the reception and discussion of its results. Firstly, it should be made clear what type of data is covered by "Baidu migration data". - are these simply inter-city trips or are they actually migrations? Migration means, at least temporarily, a change of residence.  For the sake of order, please indicate how many such migrations are contained in the dataset and what is the spread between months, what are the largest values between cities, etc. The data are the basis of the analysis and the reader needs to know a little more about them. When discussing the results and conclusions, it is essential to address the possible shortcomings of such data. Secondly, the maps presented in Fig. 4 are correct but difficult to perceive. I recommend adding some kind of synthetic map, which would indicate areas that are unchanged and those in which the belonging to "urban communities" changes. At this point, it is worth noting that the term 'communities' has a specific meaning in science, so its unorthodox use here would require some definition just like the term 'events'. Thirdly, in the conclusions/discussion of the results, there was also no reference to the applicability of such results, as mentioned in the introduction. Finally, it would have been worthwhile to at least attempt to explain such a large difference in e.g. the picture for Sept 2021 versus some average picture. In Europe or the USA, this could be related to e.g. student migration and the start of a new academic year. On a minor note, it might also be worthwhile to use the concept of proximity here and also to refer to the concept of accessibility. Congratulations on an interesting article.

Round 2

Reviewer 3 Report

Dear Authors,
Thank you for your answers and explanations. The corrections made are very beneficial to the overall perception of the article, which can be published in such a cassette.
Best regards

Author Response

We appreciate your review and comments. Your comments are helpful in revising this article.